# Artificial Intelligence (AI)-Enhanced Ultrasound Techniques Used in Non-Alcoholic Fatty Liver Disease: Are They Ready for Prime Time?

Elena Codruta Gheorghe [1], Carmen Nicolau [2], Adina Kamal [3], Anca Udristoiu [4], Lucian Gruionu [5] and Adrian Saftoiu [6,7,*]

1 Department of Family Medicine, University of Medicine and Pharmacy Craiova, 200349 Craiova, Romania; constantinescu.codruta@yahoo.com
2 Lotus Image Medical Center, ActaMedica SRL Târgu Mureș, 540084 Târgu Mureș, Romania; carmen.nicolau@gmail.com
3 Department of Internal Medicine, University of Medicine and Pharmacy Craiova, 200349 Craiova, Romania; adinaka@gmail.com
4 Faculty of Automation, Computers and Electronics, University of Craiova, 200776 Craiova, Romania
5 Faculty of Mechanics, University of Craiova, 200512 Craiova, Romania; lgruionu@gmail.com
6 Department of Gastroenterology and Hepatology, University of Medicine and Pharmacy Carol Davila Bucharest, 050474 Bucharest, Romania
7 Department of Gastroenterology, Ponderas Academic Hospital, 014142 Bucharest, Romania
* Correspondence: adriansaftoiu@gmail.com

**Abstract:** Non-alcoholic fatty liver disease (NAFLD) is the most prevalent cause of chronic liver disease, affecting approximately 2 billion individuals worldwide with a spectrum that can range from simple steatosis to cirrhosis. Typically, the diagnosis of NAFLD is based on imaging studies, but the gold standard remains liver biopsies. Hence, the use of artificial intelligence (AI) in this field, which has recently undergone rapid development in various aspects of medicine, has the potential to accurately diagnose NAFLD and steatohepatitis (NASH). This paper provides an overview of the latest research that employs AI for the diagnosis and staging of NAFLD, as well as applications for future developments in this field.

**Keywords:** liver steatosis; artificial intelligence; deep learning

## 1. Introduction

Fatty liver disease has slowly turned into a "silent pandemic" in recent years, mainly due to rising obesity and type 2 diabetes rates, with a global prevalence of approximately 25% [1]. The burden of non-alcoholic fatty liver disease (NAFLD) is, however, even greater, and its prevalence is persistently increasing at an alarming pace [2]. This is a major concern because even though NAFLD is generally considered harmless when it only involves simple steatosis, it can escalate to a more severe form known as steatohepatitis (NASH), which may lead to cirrhosis and hepatocellular carcinoma. Therefore, NAFLD is now regarded as a significant public health concern and a risk factor for higher morbidity.

Artificial intelligence (AI) is an emerging concept that refers to methods capable of performing tasks similar to human intelligence, such as learning and problem solving. Machine learning (ML) involves methods capable of analysing data and learning descriptive or predictive models. The concept of deep learning (DL) involves artificial neural networks (ANNs) that are inspired by the neural structure of the brain. Artificial neural networks or neural networks were introduced in 1944 by neuropsychologist Warren McCullough and mathematician Walter Pits. Neural networks are inspired by the neural structure of the human brain and are based on a collection of nodes called "artificial neurons" that they use to store and identify information. They are created from input and output layers, but also

from hidden layers, which are not directly visible at the input (Figure 1). These are layers of mathematical functions designed to transform input data and produce a specific output.

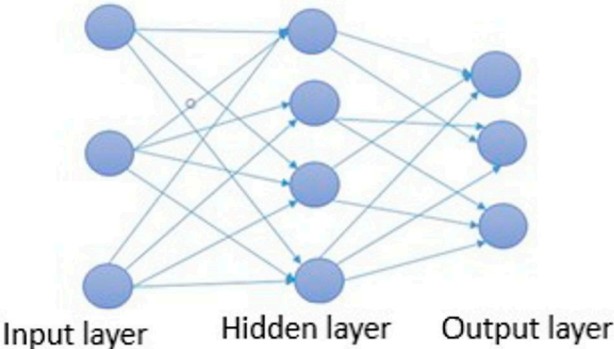

**Figure 1.** Representation of a neural network (ANN).

The human brain is organized into a complex network of about 86 billion neurons, and in turn, each neuron is typically connected to thousands of other neurons. A biological neuron consists of synapses, dendrites, a cell body and axons. In a biological neural network, dendrites receive signals from other neurons and are responsible for relaying this information to the cell body (soma). If the sum of these signals is strong enough to activate the neuron (if it reaches a certain level) then it transmits a signal along the axon, which also reaches the other neurons whose dendrites are attached to any of the axon terminals. An artificial neuron (Figure 2) is a mathematical function that simulates a biological neuron; it receives one or more inputs (x) that it processes to produce the desired result. In neural networks, neurons are connected in layers, with weights (w) relating them to their neighbouring neurons. The sum of the weights is then subjected to a non-linear function, known as a transfer function. A specific threshold value is used to compare the sum of the weights before passing through the activation function to keep the response within the desired range [3].

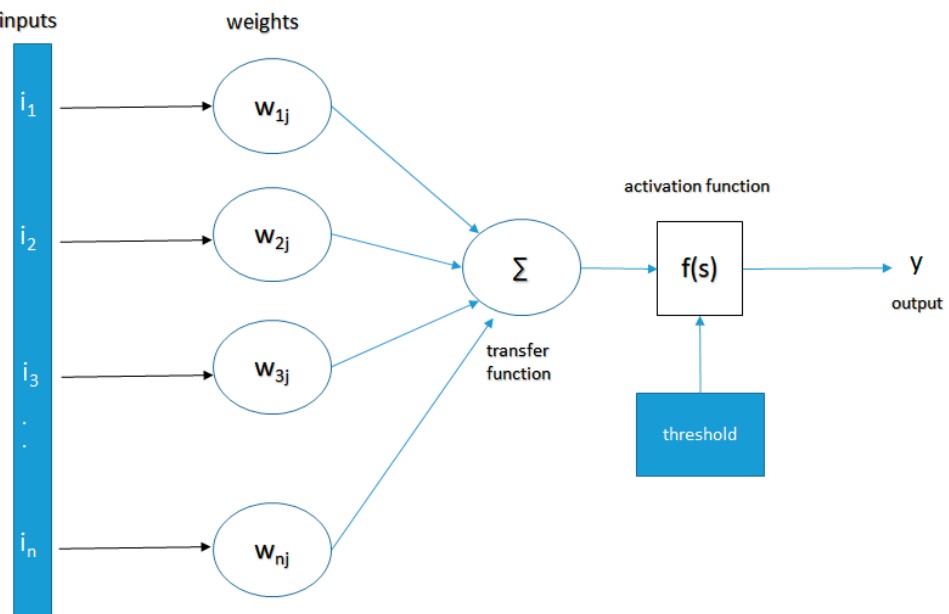

**Figure 2.** The structure of an artificial neuron.

Although AANs were the first neural networks modelled after biological neurons, (most of the time) during training, they stopped at a local minimum of the training data, leading to over-learning. Thus, neural networks have been extended to more complex

models, called deep neural networks (DNN). As shown in Figure 3, DNNs are composed of a stack of neural network layers and in order to solve a task, it is necessary to process the data from the input layers to produce an output. The higher the number of layers, the "deeper" the network is considered.

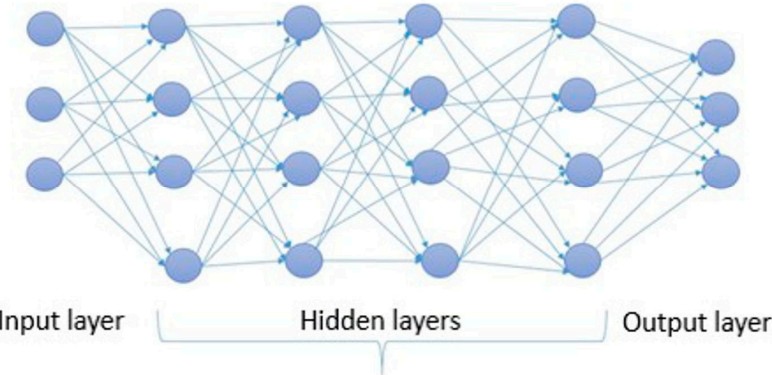

**Figure 3.** The structure of a deep neural network (DNN).

Artificial intelligence is gaining more and more ground through the excellent performance it has demonstrated in various applications, with the number of AI publications in diagnostic imaging increasing significantly in the past 15 years to more than 1000 per year. The combination of powerful hardware, new techniques and optimized libraries have enabled the development and use of convolutional neural networks (CNN) in the medical field. Although the shift from expert-based feature determination to data-driven feature learning has been gradual, DL has brought significant improvements over other ML techniques in medical image diagnosis. In particular, DL techniques have been investigated for assisting different types of diagnostic modalities (medical imaging and histopathological diagnosis) as they can help to facilitate the quantitative evaluation of imaging data in an objective and automatic way with high accuracy. The major improvements in DL over other ML techniques have had a great impact on medical image diagnosis, even though the transition from methods that use features manually determined by experts to methods that learn features from data has been gradual.

In the field of liver imaging, AI has been oriented towards various clinical applications, some of which were addressed in several studies on detection and characterisation, staging, quantifying and therapy of various liver disorders such as diffuse liver diseases and focal liver lesions (FLL) [4]. The aim of this review is to analyse the clinical impact of AI techniques for patients with NAFLD, based on recent developments in ultrasound technology.

## 2. AI Applications and Research in NAFLD

### 2.1. Liver Steatosis

NAFLD encompasses a wide range of histological conditions, from isolated steatosis to the inflammation stage, and as hepatic fibrosis progresses, it can eventually result in cirrhosis and liver cancer. In recent decades, NAFLD has made its way into the spotlight of serious liver diseases, and is now considered the most common cause of chronic liver disease [1]. The increasing prevalence of metabolic disorders such as obesity, dyslipidaemia and type 2 diabetes have made it easier to reach the actual high rates of liver steatosis. In addition to these risk factors, NAFLD has been found to be correlated with genetic factors, intestinal microbiota and diet. However, the knowledge on pathogenesis is still unclear and numerous hypotheses have been proposed. Among them, the "multiple hit" hypothesis [5] has been thoroughly investigated, suggesting that various risk factors collaborate to induce changes in the hepatocytes, also explaining why a number of patients with NAFLD do not progress to NASH or other hepatocytic lesions.

AI has been increasingly studied for liver steatosis in the past 5 years for the analysis of large amounts of images for the detection and classification of NAFLD. In 2018,

Byra et al. [6] paved the way for the automatic detection of fatty liver disease, proposing a CNN model aimed at diagnosing hepatic steatosis using B-mode ultrasound images. Their study involved 550 such images of the liver obtained from 55 obese patients who were scheduled to undergo bariatric interventions, 70% being diagnosed with hepatic steatosis. They extracted features that were compatible with steatosis and then used a DL algorithm to classify images, obtaining an overall accuracy of 96.3% compared to the hepato-renal index (HRI) and the grey level coincidence matrix (GLCM). Since then, over 50 studies have identified different AI-assisted methods for the diagnosis or staging of NAFLD.

Studies that investigate the performance of DL algorithms in diagnosing hepatic steatosis from US images are relatively heterogenous regarding the AI classifier and feature extraction. An interesting observation is that one study that used a Fuzzy neural network also used advanced image processing methods for improving the image quality, reaching an accuracy of 100%, underlying the importance of feature parameters [7]. However, all the DL models investigated have achieved a very good overall accuracy, exceeding 80% [8–13]. While the data for an automated diagnosis look promising, there is still a gap for the development of a product to meet the needs of early diagnosis. As for DL algorithms used to categorize the severity of hepatic steatosis, it appears that majority of studies used larger datasets, up to even over 30,000 ultrasound images [14]. Nevertheless, even studies with a lower amount of images analysed have achieved a similar accuracy of about 90% [15]. This may suggest that the accuracy is related to the feature extraction model rather than the AI classifier. Actually, two studies [16,17] found that ResNet had the highest accuracy compared to other CNN models and entropy imaging. While some studies have found that DL algorithms can differentiate moderate and severe steatosis more efficiently, other classified early or severe steatosis more efficiently [14,17–19]. However, all the studies concluded that AI classifiers can be reliably used for quantitative steatosis assessment, similar to or outperforming Fibroscan and the controlled attenuated parameter (CAP), showing a good correlation with MR spectroscopy and medical experts [14,15,17,19,20].

Moreover, two recent systematic reviews [21,22] addressed how AI can be integrated for the diagnosis of NAFLD, NASH and liver fibrosis, as well as how it performs on ultrasound images to diagnose and quantify NAFLD. The first review [21], also contained a meta-analysis which included 25 studies, showing an AUC of 0.98 for AI-assisted ultrasonography for the diagnosis of NAFLD, pointing out that AI-integrated methods are capable of detecting early stage steatosis, a downside of conventional ultrasound. The heterogeneity was relatively low, probably due to different diagnostic methods. As for the AI identification of NASH, the heterogeneity was higher and the AUC was lower at 0.8. The second systematic review [22] emphasised the good performance of combining AI with ultrasonography image analyses, particularly to detect early stage steatosis. Both studies confirm the superior performance of neural networks over non-neural networks [21,22].

### 2.2. Liver Fibrosis

A variable percentage of patients with NAFLD develop NASH, which furthermore leads to fibrotic changes. Sustained and progressive fibrosis is a long-term process, but patients can present with end-stage liver disease and fibrosis, which is also the most significant predictor of prognosis in NAFLD. In a systematic review and meta-analysis, Dulai et al. found that the risk for all-cause and liver-related mortality increases as the fibrosis progresses to higher stages in patients with NAFLD [23]. Thus, accurate non-invasive fibrosis tests play a significant role in lowering the increasing incidence of liver cirrhosis and hepatocellular carcinoma.

The degree of liver fibrosis is also important in the evaluation of NAFLD patients, with non-invasive methods such as real-time elastography (RTE) or shear-wave elastography (SWE) being extensively used instead of liver biopsies [24].

RTE was proposed initially for the evaluation of liver fibrosis in chronic hepatitis with a good correlation between the elasticity scores obtained using RTE and the histological fibrosis stage [25]. A direct comparison between RTE and transient elastography (Fibroscan)

showed the superiority of RTE for the diagnosis of liver fibrosis in chronic hepatitis C [26]. The method was validated in both chronic hepatitis B and C [27,28]. RTE has been used in patients with NAFLD, with a diagnostic accuracy of 82.6–96.0% based on cut-offs of 2.47 for F1, 2.67 for F2, 3.02 for F3 and 3.36 for F4 [29]. Additionally, the Liver Fibrosis Index was calculated based on RTE and showed a significant correlation with the increasing histological severity of fibrosis in both chronic hepatitis C and NAFLD [30]. These data were confirmed in a meta-analysis that compared transient elastography, acoustic radiation force impulse (ARFI) imaging and RTE, indicating a similar overall accuracy for the evaluation of significant liver fibrosis [31]. Real-time SWE is also considered accurate for the evaluation of liver fibrosis in chronic hepatitis C, more accurate than transient elastography for the assessment of significant liver fibrosis [32]. A recent meta-analysis showed that pSWE and transient elastography have similar accuracies in the detection of significant fibrosis, advanced fibrosis and cirrhosis in patients with NAFLD [33]. Even though both methods have limitations for obese patients, 2D SWE was shown to be feasible in severely obese patients, as it was successfully performed in 97.3% of patients and was correlated with BMI, waist circumference, NAFLD activity score and steatosis in a univariate analysis [34]. Furthermore, transient elastography, 2D SWE and MRI have similar diagnostic accuracies for significant and especially for advanced fibrosis in patients with biopsy-proven NAFLD [35]. Multiparametric ultrasound techniques including dispersion slope ((m/s)/kHz), attenuation coefficient (dB/cm/MHz) and shear-wave speed (in meters per second) allow the discrimination of inflammation (A1 to A3), steatosis (S1 to S3) and fibrosis (F1 to F4: cirrhosis), with the combination yielding an area under the receiver operating curve of 0.81% for patients with biopsy-proven NAFLD [36]. The introduction of attenuation imaging represented a significant step forward in ultrasound technologies in order to accurately quantify liver fibrosis and steatosis in NAFLD patients [37]. The accuracy of various techniques of elastography (vibration controlled transient elastography, pSWE and 2D SWE) in comparison with MRI in NAFLD patients was analysed in a recent meta-analysis, showing acceptable summary estimates of the area under the receiver operating curve for the diagnosis of significant fibrosis, advanced fibrosis and cirrhosis [38]. The multitude of ultrasound elastography methods used for the identification of advanced liver fibrosis in patients with NAFLD and NASH clearly allow the quantitative characterisation of inflammation, steatosis and fibrosis, but a multistep algorithmic approach is certainly needed, with AI techniques being needed for distinct clinical decision-making scenarios [39]. The application of non-invasive ultrasound elastography techniques such as SWE is especially important for identifying the presence of significant fibrosis in paediatric patients, where liver biopsy should be avoided [40].

Various AI and ML approaches have already been used for the evaluation of liver fibrosis based on either RTE or SWE, as both methods are still difficult to interpret in a clinical setting [41,42]. Thus, various ML classifiers (i.e., support vector machine, naïve Bayes, random forest and K-nearest neighbour) have been employed to stage liver fibrosis in chronic B hepatitis, outperforming the original Liver Fibrosis Index developed through regression analyses based on RTE images [41]. Another stiffness value clustering and ML algorithm indicated that the classification of healthy versus chronic liver disease patients based on a support vector machine (SVM) model had 87.3% accuracy with sensitivity and specificity values of 93.5% and 81.2%, respectively [42]. Additionally, DL has been used to assess liver fibrosis in a recent prospective multicentric trial, showing high values of the area under the receiver curve of 0.97 (for F4), 0.98 (for ≥F3) and 0.85 (for ≥F2) [43]. Based on 398 patients and 1990 images included, this approach yielded a better overall performance in predicting liver fibrosis stages compared with 2D-SWE and biomarkers, at least for the patients with chronic HBV infection. Nevertheless, both RTE and SWE still have a significant bias with intra- and inter-observer variability, whilst automated frameworks such as SWE-Assist can help by checking SWE image quality, selecting a region of interest and classifying the ROI into fibrosis stages (over F2) [44]. This kind of automated AI and DL/ML analysis based on CNNs will certainly help doctors to easily classify and

stage inflammation, steatosis and fibrosis based on non-invasive ultrasound techniques that are highly accurate, extremely cheap and largely available in the primary setting or at the point-of-care. A recent meta-analysis studied AI-assisted ultrasonography and elastography, but also CT, MRI and clinical parameters, showing high sensitivity, specificity, PPV, NPV and diagnostic odd ratios (DORs), especially for liver steatosis, but also for significant fibrosis, advanced fibrosis and liver cirrhosis [45].

### 2.3. Liver Cirrhosis

For a long time, the leading cause of cirrhosis worldwide was viral hepatitis, but NASH and alcoholic fatty liver disease (AFLD) are soon expected to surpass it as the primary causes of cirrhosis [46,47]. Currently, the prevalence of NAFLD-associated cirrhosis in the United States is relatively low, at around 1–2% [48], but screening for chronic liver disease in NAFLD high-risk groups is a key strategy for HCC surveillance programs.

The importance of elastography for the early identification of patients with liver cirrhosis has been highlighted in a clinical guideline accompanied by a technical review of the American Gastroenterology Association (AGA) [49,50]. Thus, non-invasive imaging modalities for the evaluation of chronic liver diseases (chronic hepatitis B and/or C, NAFLD or AFLD) prioritised VCTE and MRI elastography looking at (1) the diagnostic performance of VCTE and MRE relative to non-proprietary, serum-based fibrosis markers for the detection of cirrhosis in patients with chronic liver disease mentioned above; (2) the performance of specific VCTE-defined liver stiffness cut-offs as a test replacement strategy for liver biopsies in establishing clinical decision-making algorithms for these patients; and (3) the performance of specific VCTE-defined liver stiffness cut-offs as a triage test to identify patients with a low likelihood of high-risk oesophageal varices (EVs) or having clinically significant portal hypertension (for pre-surgical risk stratification). Meanwhile, other non-invasive modalities for the assessment of fibrosis (e.g., RTE, ARFI or pSWE/2D-SWE) or steatosis (CAP or MRI—estimated proton density fat fraction) have been developed and will certainly be used in clinical decision-making practice algorithms [51]. Thus, according to the latest data, a "rule of five" for LSMs with VCTE and a "rule of four" for LSMs with the ARFI-based techniques have been proposed to discriminate various fibrosis stages. Furthermore, in patients with advanced CLD, the risk of liver decompensation increases with an increasing liver stiffness value, whilst SWE has been proposed as a risk predictor of morbidity and mortality in patients with cirrhosis. Both VCTE and ARFI techniques have already been validated as non-invasive methods for the screening of varices in this setting. Even more, LSM measurements were part of predictive algorithms for HCC occurrence in this high-risk population with advanced chronic liver diseases. For clinical validation, the AGA studied patients with NAFLD and predicted advanced liver fibrosis [52]. Thus, patients aged over 18 with a Fibrosis-4 (FIB-4) score of less than 1.3 and an LSM of <8 kilopascals (kPa) by VCTE have a low risk of significant fibrosis, although other strategies were deemed necessary to predict patients with advanced fibrosis.

Based on all these endeavours and the plethora of non-invasive markers used for risk prediction in chronic liver diseases, it is obvious that AI and DL/ML techniques will be the future avenue for assessment of these patients. Conventional approaches based on ANNs and parenchymal echo patterns of greyscale imaging were already employed even 25 years ago in order to better assess chronic hepatitis and regenerative nodule characteristics of liver cirrhosis [53]. The same group employed a similar methodology to assess the risk of HCC in liver cirrhosis patients [54,55]. The next step was to analyse the Doppler information yielded by colour flow and pulsed Doppler measurements using an ANN [56]. Thus, several US parameters (liver parenchyma, thickness of spleen, hepatic vein waveform, hepatic artery pulsatile index and hepatic vein damping index) were used to establish an ANN model capable of quantifying advanced liver fibrosis or cirrhosis. ANNs including several serological tests (either alone or combined into scores) and liver stiffness measurements based on VCTE were further analysed for the diagnosis of cirrhosis

and significant portal hypertension and oesophageal varices (with an estimated accuracy of over 80%) [57].

DL approaches have developed over time and have led to the automatic classification of liver fibrosis, in comparison with non-deep learning-based algorithms (artificial neural networks, multinomial logistic regression, support vector machines and random forests) [58]. CNNs used in this particular study lead to higher areas under receiver operating characteristic curve, with values of up to 0.85–0.95 when automatically scoring liver fibrosis stages. Another study focused on the prediction of the METAVIR score based on a –four-class model (F0 vs. F1 vs. F2–3 vs. F4) based on 3446 patients and 13,608 US images, validated on a separate internal set (266 patients and 300 images) and an external set (572 patients and 1232 US images) [59]. The areas under the receiver operating curve (AUROC) for classification of liver cirrhosis were 0.901 (internal test set) and 0.857 (external test set). Moreover, the AUROC for the DCNN classification of liver cirrhosis based on US imaging was significantly higher than all five radiologists, at least for the prediction of the METAVIR score.

ML is useful to predict bleeding oesophageal varices in compensated advanced chronic liver disease, based on laboratory measurements and liver stiffness measurements used to generate an AI algorithm based on extreme-gradient boosting [60]. The study included 828 patients with advanced chronic liver diseases and oesophageal varices, mostly with NAFLD, AFLD and hepatitis B and C, with the DL-assisted algorithm exhibiting an accuracy of 98.7% in predicting variceal bleeding, better than endoscopic classification alone, which reached only 58.9%. A similar multicentric approach (17 institutions from China, Singapore and India) used an ML-based strategy to predict the presence of high-risk varices in compensated cirrhosis to avoid unnecessary endoscopies based on liver stiffness, platelet count and total bilirubin [61]. The model worked well for the prediction of high-risk varices, and is also accompanied by a web-based calculator (http://www.pan-chess.cn/calculator/MLEGD_score, accessed on 4 March 2023). Thus, the ML-based model spared 52.6% of unnecessary endoscopies (in the training cohort) with a missed high-risk varices rate of 3.6%, and spared 58.1% (in the validation cohort) with a missed high-risk varices rate of 1.4%. Similar numbers were also obtained based on two separate external test cohorts.

Another ML model was used to predict portal vein thrombosis (PVT) after splenectomy in portal hypertension patients, with a good accuracy and a satisfactory agreement between prediction models and real life observations [62].

### 2.4. Liver Tumours

Approximately 25% of patients with NAFLD may develop steatohepatitis (NASH), which is further associated with other complications of liver cirrhosis (hepatic encephalopathy, ascites and hepato-renal syndrome, upper gastrointestinal bleeding, etc.) and development of HCC. The prognosis of HCC is considered dismal as it represents the 2nd leading cause of cancer-related death worldwide. A particular feature of NAFLD-associated HCC is that it involves a higher risk of non-cirrhotic HCC compared to other chronic liver diseases [63], accounting for about 25–45% of the total cases of HCC in NAFLD patients [64]. The epidemiology for NAFLD-associated HCC is similar to NAFLD cirrhosis, with data from the first decade of this century indicating that NASH has become the second most common cause of HCC leading to liver transplants in Unites States [65].

A recent clinical practice update from the American Gastroenterological Association addressed the topic of surveillance of NAFLD for HCC, including optimal screening tools, frequency of monitoring and the presence of risk factors [66]. An interesting approach was to develop a SWOT (strengths, weaknesses, opportunities and threats) analysis in order to identify non-invasive tests used in NAFLD to select high-risk patients for NASH. The facilitates the surveillance of HCC in order for early diagnosis and to possibly treat and assess the effectiveness of interventions [67].

An initial systematic review focused on medical imaging and artificial intelligence identified 11 initial papers that studied the differential diagnosis of focal liver lesions (FLL),

with the aim of diagnosing, segmenting or differentiating HCC based on various CNN techniques [68]. Whilst conventional ANNs were scarcely used initially [69–71], it soon became clear that a more structured approach using DL and CNNs would be needed for automatic recognition of HCC areas in ultrasound images [72].

A recent review analysed the value of AI and DL for the diagnosis, prognosis and therapy of HCC [73]. Newer techniques integrate sets of textured features into ensemble models further analysed by CNN to differentiate normal liver, chronic hepatitis, liver cirrhosis and HCC with an overall accuracy of 96.6% [74]. Furthermore, real-time systems based on full-length ultrasound movies have recently been developed, achieving an overall detection rate of 89.8%, significantly higher than that achieved by non-radiologist physicians and radiologists [75].

Nevertheless, for correct assessment, contrast-enhanced ultrasound (CEUS) should be employed, as this has better accuracy than greyscale ultrasound imaging alone [76]. Quantitative analysis approaches have been developed recently to avoid errors induced by motion artifacts and the dynamic changes in contrast enhancement during different vascular phases (arterial, portal and venous), achieving an accuracy of 0.84 in the distinction between benign and malignant phases [77]. Similar DL approaches are useful for the differential diagnosis of focal nodular hyperplasia and atypical HCC [78] or for the differential diagnosis of HCC and intrahepatic cholangiocarcinoma [79]. Last, but not least, a similar approach has been employed in the evaluation of patients after therapy, either after transarterial chemoembolization, microwave/radiofrequency ablation or surgery [80,81].

## 3. Discussion

For the automatic diagnosis of liver steatosis, out of the algorithms discussed, CNNs exhibit superior performance compared to more basic logistic regression models. AI models can accurately detect early stage steatosis on a level similar to or outperforming other non-invasive techniques (Fibroscan and CAP). These models could thus be used as a screening tool for identifying patients with NAFLD, even in early stages, in the general population. In addition, the utilization of CNN classification algorithms has the potential to replace liver biopsies for assessing the severity of NAFLD and liver fibrosis in certain patients. CNNs have successfully graded F3 and F4 fibrosis using 2D SWE, surpassing the performance of only 2D-SWE and biomarkers, proving it can be the current standard for classification of advanced fibrosis and cirrhosis. CNNs also demonstrate exceptional performance in the classification of liver tumours (detection, characterization and segmentation). Even though real-time systems achieve a high detection rate, for the moment, AI can only be regarded as an assistant for image interpretation.

Moreover, another area where DL can assist clinicians in quick and accurate decision making is in AI automated ultrasound report generation. Usually, the process of writing comprehensive reports on the ultrasound assessment can be cumbersome, time consuming and prone to error. Thus, DL can answer this need by image caption algorithms, which involves computers generating captions by processing visual inputs such as images [82,83]. An interesting approach will be to use generative pre-trained transformer (GPT)-based models, which have suggested in a variety of applications for report generation, educational support, clinical decision support, patient communication and data analysis [84]. For example, ChatGPT is a variant of GPT that is fine-tuned for conversational language understanding and generation, possibly useful for radiologists and ultrasonographers, with various advantages but also limitations.

Undoubtedly, AI has generated extensive discussions with great excitement, but there are still limitations and obstacles to be addressed and to take into consideration in future research in this area. These include various different diagnostic methods among studies, the need for large and standardized image databases, the use of different AI classifiers in research, implementation of classification models in hardware for clinical use and overdiagnosis by detecting minor changes that could indicate indolent disease.

## 4. Conclusions

Data from medical imaging research support the implementation of AI in healthcare settings, with clinical implications such as improving the performance of early stage NAFLD diagnosis, accurately quantifying NAFLD and estimating the stage of liver fibrosis, decreasing subjectivity and minimizing human errors, as well as introducing computer-aided diagnosis for less experienced doctors.

Thus, in the future, considering all the presented research, AI techniques integrated with ultrasound could support clinical decision making in the management of NAFLD patients. However, despite the increasing interest in this area of research, currently there are only a limited number of clinically approved and available applications for diagnosis and prognosis of NAFLD. The next step in implementing AI-assisted models in clinical practice is to extend the research in larger studies with external and prospective validation and to improve the healthcare infrastructure.

**Funding:** This research was funded by a grant from the Ministry of Research and Innovation, grant number ID P_34_498, within MFE 2014-2020-POC.

**Institutional Review Board Statement:** This study did not require ethical approval.

**Informed Consent Statement:** Not applicable.

**Data Availability Statement:** No new data were created in this article. Data sharing is not applicable to this article.

**Conflicts of Interest:** The authors declare no conflict of interest.

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
