# Peer review of "Artificial Intelligence (AI)-Enhanced Ultrasound Techniques Used in Non-Alcoholic Fatty Liver Disease: Are They Ready for Prime Time?"

_applsci, doi:10.3390/app13085080_

Round 1

Reviewer 1 Report

Elena Codruța Gheorghe et al. wrote a review regarding "artificial Intelligence (AI) Enhanced Ultrasound Techniques Used in Liver 2 Steatosis: Are They Ready for Prime Time?". 

This work looks interesting i think that the idea of the manuscript is good. I have no comments.

good luck for all

Author Response

We are grateful for dedicating your time to evaluate the manuscript. 

Best regards, 

Reviewer 2 Report

MS (applies-2318063) aims to analyze the clinical impact of DL for patients with liver steatosis, based on newer developments in ultrasound technology. For this purpose, in this study, Liver steatosis, Liver fibrosis, Liver cirrhosis, Liver tumors, and reporting applications were examined.

-Liver steatosis, Liver fibrosis, Liver cirrhosis, Liver tumors, and reporting are given under separate headings. These should be gathered under a title and given as a subheading.

-The title of Figure 2 should be given under the figure. Also, the shape is blurry and the text size is disproportionately large. It must be redrawn.

- Punctuation should be corrected in line 64.

- Line 281, "(F0 vs. F1 vs. F23 vs. F4)" check the "F23" is it F2-3?

- MS needs a discussion part.

- I think it would be better to remove the "reporting section" and discuss this information in the discussion section.

Author Response

Point-by-point Response to Reviewer’s Comments

We appreciate the time and effort that you and the reviewers have dedicated to providing your valuable feedback on my manuscript. Please see my point-by-point response below to the comments and concerns.

- Liver steatosis, Liver fibrosis, Liver cirrhosis, Liver tumors, and reporting are given under separate headings. These should be gathered under a title and given as a subheading.

Answer: We have gathered them all under a common title “AI applications and research in NAFLD” and each section as a subheading.

-The title of Figure 2 should be given under the figure. Also, the shape is blurry and the text size is disproportionately large. It must be redrawn.

Answer: We replaced the image with another one with higher quality and moved the title under the figure. 

- Punctuation should be corrected in line 64.

Answer: This has been addressed.

- Line 281, "(F0 vs. F1 vs. F23 vs. F4)" check the "F23" is it F2-3?

Answer: This has been addressed accordingly (F2-F3).

- MS needs a discussion part.

Answer: This has been addressed and there is now a separate Discussion part.

- I think it would be better to remove the "reporting section" and discuss this information in the discussion section.

Answer: This has been addressed and the “reporting section” has been moved to “Discussion”

Reviewer 3 Report

Dear colleagues!

I read with interest your manuscript entitled "Artificial Intelligence (AI) Enhanced Ultrasound Techniques Used in Liver Steatosis: Are They Ready for Prime Time?", which is a structured/narrative review. The subject is actual and may be interesting for the readers. I have only a few minor comments that, being addressed, could help you to make the manuscript even better. 

1. The title emphasizes the use of ultrasound techniques in liver steatosis. However, significant part of the paper discusses the use of AI-driven US techniques for detection of liver fibrosis, cirrhosis and even tumors. Isn't it better to make changes to the title (for example, to use "pathophysiological conditions of the liver" or "liver disorders" instead of "steatosis")? This would be closer to what is describe. Otherwise, there would be a need to shorten the body of the paper, which is less preferrable. 

2. The aim of the study might require similar changes. Please, check. 

3. The paper lacks Discussion section, but contains 2 chapters with conclusions (7&8).  Please, check for possible misprint and ensure that the results are discussed appropriately. 

4. It seems that information in both conclusion sections is quite vague. Is it possible to make certain conclusions based on what is described (current state of the problem for each topic, unmet needs, further research direction)?

5. Image quality for figure 2 is not optimal, please, make the resolution higher.

6. Figure 2 requires legend: please, disclose abbreviations here. 

7. There are some misprints throughout the text, although they don't intervene the understanding. Please, check and revise

Author Response

Point-by-point Response to Reviewer’s Comments

We are grateful to the reviewers for their insightful comments on our paper. We have been able to incorporate changes to reflect most of the suggestions provided by the reviewers. Please see my point-by-point response below to the comments and concerns.

  1. The title emphasizes the use of ultrasound techniques in liver steatosis. However, significant part of the paper discusses the use of AI-driven US techniques for detection of liver fibrosis, cirrhosis and even tumors. Isn't it better to make changes to the title (for example, to use "pathophysiological conditions of the liver" or "liver disorders" instead of "steatosis")? This would be closer to what is describe. Otherwise, there would be a need to shorten the body of the paper, which is less preferrable. 

Answer: We have changed the title to “Artificial Intelligence (AI) Enhanced Ultrasound Techniques Used in Non-alcoholic Fatty Liver Disease: Are They Ready for Prime Time?” as our paper mainly focuses on the use of AI in the diagnosis and staging of NAFLD and its complications.

  1. The aim of the study might require similar changes. Please, check. 

Answer: We have changed it to “The aim of this review is to analyse the clinical impact of AI techniques for the patients with NAFLD, based on newer developments in ultrasound technology.” – having the same reasoning as above mentioned. 

  1. The paper lacks Discussion section, but contains 2 chapters with conclusions (7&8).  Please, check for possible misprint and ensure that the results are discussed appropriately. 

Answer: This has been addressed and there is now a separate Discussion part.

  1. It seems that information in both conclusion sections is quite vague. Is it possible to make certain conclusions based on what is described (current state of the problem for each topic, unmet needs, further research direction)?

Answer: This has been addressed and we have now some more specific conclusions.

  1. Image quality for figure 2 is not optimal, please, make the resolution higher.

Answer: We replaced the image with another one with higher quality.

  1. Figure 2 requires legend: please, disclose abbreviations here.

Answer: Figure 2 has a legend and has been moved under the figure.  

  1. There are some misprints throughout the text, although they don't intervene the understanding. Please, check and revise

Answer: This has been addressed and we have corrected misprints and typing mistakes as suggested.